# Electrospinning and Post-Spun Chain Conformations of Synthetic, Hydrophobic Poly(*α*-amino acid)s

**DOI:** 10.3390/polym12020327

**Published:** 2020-02-04

**Authors:** Kesavan Devarayan, Souta Nakagami, Shuichi Suzuki, Ichiro Yuki, Kousaku Ohkawa

**Affiliations:** 1Department of Basic Sciences, College of Fisheries Engineering, Tamil Nadu J. Jayalalithaa Fisheries University, Vettar River View Campus, Nagapattinam 611 002, India; kesavannitt@gmail.com; 2Division of Synthetic Polymers, Institute of High Polymer Research, Faculty of Textile Science and Technology, Shinshu University, Tokida 3-15-1, Ueda 386-8567, Japan; s-nakagami@takiron-ci.co.jp; 3Department of Neurological Surgery, University of California, Irvine, 200S Manchester St., Suite 210, Orange, CA 92868, USA; shuichis@hs.uci.edu (S.S.); iyuki@hs.uci.edu (I.Y.); 4Department of Neurosurgery, The Jikei Univeristy Hospital, 3 Chome-25-8 Nishishinbashi, Minato City, Tokyo 105-8461, Japan; 5Division of Biological and Medical Fibers, Institute for Fiber Engineering, Interdisciplinary Cluster for Cutting Edge Research, Shinshu University, Tokida 3-15-1, Ueda 386-8567, Japan

**Keywords:** poly(*α*-amino acid)s, synthesis, electrospinning, conformational analysis, molecular weight

## Abstract

Electrospinning and post-spun conformations of hydrophobic poly(*α*-amino acid)s are described in this study. The poly(*α*-amino acid)s, poly(Gly), poly(l-Ala), poly(l-Val), and poly(l-Leu) were synthesized via corresponding *N*-carboxy-*α*-amino acid anhydrides. The average molecular weight and degree of polymerization of these polymers were determined by *N*-terminus labeling using 2,4-dinitrofluorobenzene and by viscometry in the case of poly(Gly). These poly(*α*-amino acid)s were electrospun from trifluoroacetic acid or trifluoroacetic acid/dichloromethane solutions. The FT-IR spectroscopy and wide-angle X-ray diffraction indicated that the electrospun poly(l-Ala) and poly(l-Leu) fibers predominantly adopts *α*-helical structure, whereas poly(l-Val) and poly(Gly) fibers exhibited mainly *β*-strand and random coil structures, respectively.

## 1. Introduction

Electrospinning is a state-of-the art technology for the production of fibers from a broad spectrum of polymers including synthetic polymers [1], polysaccharides [2,3,4], proteins [5,6,7], and DNA [8]. The parameters for successful electrospinning include applied voltage, tip-to-collector distance, solvent, conductivity, solution concentration, and viscosity. The electrospun fibers are often collected as the sheet like non-woven (ESNW) fabrics. The ESNWs are potential materials for applications such as protective textiles [1,9], scaffolds for tissue engineering [10], air filters [11,12], actuators [13], chemosensors for detection of metal ions [14], and so on [1,15]. The differences in the fiber properties are mainly due to the chain conformation of the polymers with respect to the spinning conditions. Therefore, control on the post-spun structure of the polymer molecular chains is important for better fiber performances. 

Synthetic poly(amino acid)s have a backbone structure identical to the natural proteins that adopts both ordered and/or disordered structures. Thus poly(amino acid)s are key models for understanding the protein folding as well as the bio-related materials science. Our research group examined the chain conformations of the poly(*γ*-benzyl-l-glutamate) (PBLG) in pre- and post-electrospinning process [15]. The PBLG was electrospun in dichloromethane (DCM) and DCM/trifluoroacetic acid (TFA). Even as the pre-spun solution in TFA or DCM/TFA exhibited random chain conformation, the molecular chains in post-spun PBLG fibers showed *α*-helical structure. There were very few succeeding studies so far on electrospinning and post-spun chain conformations of poly(*α*-amino acid)s and polypeptides. For example, poly(l-ornithine) [16] and poly(l-glycine-*co*-l-tyrosine) [17] in deionized water were electrospun to give fibers having average fiber diameter at micron-scale. The molecular chains of the post-spun fibers of poly(l-glycine-*co*-l-tyrosine) were predominantly *β*-sheet structure [18]. In a recent study, poly(l-Phe) in hexane/chloroform was electrospun and demonstrated as a superhydrophobic material [19]. However, studies on synthetic hydrophobic poly(*α*-amino acids) such as poly(Gly), poly(l-Ala), poly(l-Val), and poly(l-Leu) for electrospinning and post-spun chain conformations are not reported.

Therefore, this study was carried out with encompassing the following objectives; (i) synthesis of hydrophobic poly(*α*-amino acids) and determination of molecular weight, (ii) electrospinning of poly(*α*-amino acids), and (iii) chain conformation analysis of the post-spun fibers. To understand the behavior of molecular chains, cast film of poly(l-Ala) was also prepared as a comparison to the poly(l-Ala)-ESNW and analyzed by means of FT-IR and WAXD.

## 2. Experimental Section

### 2.1. Materials

*N^α^*-benzyloxycarbonyl-l-alanine (Z-l-Ala), glycine (Gly), phosphorous pentachloride (PCl_5_), bis(trichloromethyl)carbonate (TCMC), triethyl amine (TEA), cyclohexyl amine (CHA), dichloromethane (DCM), dinitrofluoro benzene (DNFB), and benzyloxycarbonyl chloride were purchased from Wako Pure Chemical Industries, Ltd., Osaka, Japan. l-Valine (l-Val) and l-leucine (l-Leu) were purchased from Nippon Rikagakuyakuhin Co., Ltd., Japan. Z-Gly was synthesized by a method described by Bergmann et al. [20]. 1-butyl-2-methylimidazolium hexafluorophosphate ([BMIM][PF_6_]) and trifluoroacetic acid (TFA) were purchased from Tokyo Chemical Industries, Co., Ltd., Japan, while 1-butyl-2-methyl-imidazolium chloride ([BMIM][Cl]) was obtained from Sigma-Aldrich, Japan (Tokyo).

### 2.2. Synthesis of Poly(α-amino acids)

#### 2.2.1. General Procedure for Synthesis of Poly(*α*-amino acids)

At first, the *N*-carboxy-*α*-amino acid anhydrides (NCA) of Z-l-Ala, l-Val, l-Leu, and Z-Gly were prepared using TCMC or PCl_5_ [21]. The NCAs were dissolved in anhydrous benzene, 1,4-dioxane, or in ionic liquid/DMSO solvent mixture and then the polymerization was initiated by TEA or by ionic liquids in the case of Gly NCA. The polycondensation reactions were monitored by FT-IR spectroscopy for the completion of polymerization. The product was precipitated using methanol or diethyl ether and the precipitate was recovered by filtration. The molecular weight and degree of polymerization of these poly(*α*-amino acids) were determined as described in the Appendix A.

#### 2.2.2. l-Ala NCA (**1**)

To a solution of Z-l-Ala (26.8 g, 120 mmol) in anhydrous ethyl acetate (80 mL), PCl_5_ (25.0 g, 120 mmol) was added on an ice bath. The reaction mixture was concentrated under reduced pressure and the product was crystallized using anhydrous hexane. The crude crystals were re-crystallized twice from anhydrous 1,4-dioxane/hexane. Then, the product was dissolved in 1,4-dioxane and passed through a column packed with activated charcoal. The eluate was condensed and crystallized by anhydrous hexane to give white crystalline product, l-Ala NCA (**1**). Yield: 10.8 g, 78.2%. IR: 1751 cm^−1^ and 1833 cm^−1^ (*ν*_C=O_, cyclic anhydride). 

#### 2.2.3. Poly(l-Ala) (**2**)

l-Ala NCA (**1**) (2.00 g, 17.4 mmol) was dissolved in 20 mL of anhydrous 1,4-dioxane. A 0.08 eq.mol of TEA was added to the solution with stirring. The reaction mixture was kept stirring for 1 week at room temperature and then heated to 80 °C to terminate the reaction. The product poly(l-Ala) (**2**) was precipitated using diethyl ether and the crystals were collected by filtration. Yield: 1.07 g, 86.8%. IR: 1654 cm^−1^ (amide I) and 1548 cm^−1^ (amide II). 

#### 2.2.4. l-Val NCA (**3**)

l-Val (9.00 g, 76.8 mmol) was dissolved in 90 mL of anhydrous 1,4-dioxane; 0.45 g of dry activated charcoal was added. TCMC (8.36 g, 28.1 mmol) was added to the reaction mixture and stirred for 90 min at 65 °C. Then, the charcoal was removed by filtration. The filtrate was condensed under vacuo and crystallized using anhydrous hexane. The crude product was recrystallized twice from anhydrous 1,4-dioxane/hexane and was purified by a procedure similar to L-Ala NCA (**1**) to give white crystalline product, L-Val NCA (**3**). Yield: 8.00 g, 72.8%. IR: 1837 cm^−1^ and 1751 cm^−1^ (*ν*_C=O_ of cyclic anhydride).

#### 2.2.5. Poly(l-Val) (**4**)

The polymerization of l-Val NCA (**3**) (3.00 g, 21.0 mmol) in anhydrous benzene (30 mL) was initiated by TEA (0.02 eq.mol). The reaction mixture was stirred for 7 days at room temperature followed by termination at 80 °C. Then, the product poly(l-Val) (**4**) was precipitated using diethyl ether. Yield: 1.53 g, 73.3%. IR: 1637 cm^−1^ (amide I) and 1540 cm^−1^ (amide II).

#### 2.2.6. l-Leu NCA (**5**)

To the solution of l-Leu (9.00 g, 68.6 mmol) dissolved in 90 mL of anhydrous 1,4-dioxane, dry charcoal (0.45 g) and TCMC (7.45 g, 25.2 mmol) were added and the reaction mixture was stirred at 65 °C for 90 min. The charcoal was filtered off and the product was recovered by a similar procedure to compound **3**. Yield: 6.08 g, 56.4%. IR: 1855 cm^−1^ and 1758 cm^−1^ (*ν*_C=O_ cyclic anhydride).

#### 2.2.7. Poly(l-Leu) (**6**)

Similar to **3**, l-Leu NCA (**5**) (3.00 g, 19.3 mmol) was polymerized in anhydrous benzene (30 mL). Yield: 1.77 g, 80.8%. IR: 1652 cm^−1^ (amide I) and 1544 cm^−1^ (amide II).

#### 2.2.8. Gly NCA (**7**)

Z-Gly (15.0 g, 71.7 mmol) was dissolved in anhydrous ethyl acetate (150 mL) and added PCl_5_ (14.9 g, 71.7 mmol) on an ice bath. The product was obtained by a similar procedure to compound **1**. Yield: 6.07 g, 83.8%. IR: 1867 cm^−1^ and 1760 cm^−1^ (*ν*_C=O_ of cyclic anhydride).

#### 2.2.9. Poly(Gly) (**8**)

GlyNCA (**7**) (6.07 g, 60.1 mmol) was polymerized in a mixture of ionic liquids/solvent, [BMIM][PF6] (5.00 g) and [BMIM][Cl] (5.00 g) and anhydrous dioxane (5 mL) for 21 h at room temperature. The product poly(Gly) was precipitated using methanol. Yield: 3.00 g, 87.5%. IR: 1647 cm^−1^ (amide I) and 1559 cm^−1^ (amide II).

### 2.3. Electrospinning of Poly(amino acids)

The pre-spun solutions of poly(l-Ala) (**2**), poly(l-Val) (**4**), poly(l-Leu) (**6**), and poly(Gly) (**8**) were prepared in TFA and TFA-DCM solvent mixture at the concentration of 4–25 wt.%. The solvents TFA and DCM were mixed at the weight ratios TFA:DCM = 10:0, 8:2, 5:5, and 2:8. The electrospinning experimental set up was assembled similar to our previous reports [3,4,22]. The pre-spun solutions of each poly(amino acid)s were separately placed into a 1-mL syringe with a tip having 0.6 mm inner diameter. A copper wire was inserted into the pre-spun solution, which was connected to the positive electrode. A copper plate covered with aluminum foil was grounded to earth and used as a collector. The electric field was generated by a high voltage power supply (HAR-50p2, Matsusada Precision Inc., Japan). Electrospinning was carried out at 14–16 kV. The tip to collector distance was 15 cm.

### 2.4. Preparation of Poly(l-Ala) Film

Thirteen milligrams of poly(l-Ala) (**2**) was first dissolved in 65 µL of methanesulfonic acid (MSA) (Sigma), then the volume was made up to 750 µL using TFA. Then, the solution was cast in a glass Petri dish by hot air to give transparent film of poly(l-Ala). To remove the remaining MSA and TFA, the film was washed with diethyl ether and then dried in vacuo at room temperature. Approximately, 1 cm^2^ of the film was cut and drawn axially up to 400% of its original length. The film was kept wet by dropping diethyl ether onto the film during elongation.

### 2.5. Wide-Angle X-ray Diffraction (WAXD)

The ESNWs were molded into pellets for WAXD measurement. The instrument was set at 40 kV, 150 mA current and a light source having wavelength of 1.54 Å was used. The diffraction intensity was measured using a Rigaku Geiger flex instrument (Tokyo, Japan).

### 2.6. FT-IR Spectroscopy

The ESNWs of poly(amino acids), except poly(Gly), were subjected to FT-IR spectral measurement in transmission mode at 400 to 4000 cm^−1^ frequency range. As for the poly(Gly)-ESNW, the attenuated total reflection (ATR) technique has been employed using a KRS5 ATR prism, as peeling-off of the poly(Gly)-ESNW from the aluminum foil was difficult.

## 3. Results and Discussion

### 3.1. Synthesis of Poly(amino acids)

Scheme 1 shows the synthesis of poly(l-Ala) (**2**), poly(l-Val) (**4**), poly(l-Leu) (**6**), and poly(Gly) (**8**). At first, the reactive NCAs were synthesized with purified yields 57–84% and characterized by FT-IR spectroscopy. PCl_5_ was used to prepare l-Ala NCA (**1**) and l-Gly NCA (**7**). TCMC was used to prepare l-Val NCA (**3**) and l-Leu NCA (**5**). The polycondensation was initiated by triethyl amine to give poly(l-Ala) (**2**), poly(l-Val) (**4**), and poly(l-Leu) (**6**). Polymerization of Gly NCA (**7**) was initiated by the mixture of ionic liquids: [BMIM][PF_6_] and [BMIM][Cl]. The poly(amino acid)s were synthesized with yields 73–88% and were confirmed by FT-IR spectroscopy. Recently, Endo et al. [23] first reported polymerization of *γ*-benzyl-l-glutamate NCA in ionic liquids using *n*-butylamine as an initiator. In this study, poly(Gly) was synthesized in ionic liquids by an amine initiator-free method. 

On one hand, the poly(l-Ala) (**2**), poly(l-Val) (**4**), and poly (l-Leu) (**6**) were not soluble in dichloroacetic acid for conventional viscometric studies. Therefore, the number average molecular weight (*M_n_*) and degree of polymerization (*Dp*) of these poly(amino acid)s were determined by *N*-terminus labeling and the values were given in Appendix A. The average *Dp* values for poly(l-Ala) (**2**) was estimated as *Dp^N^* = 2570; for poly(l-Val) (**4**), *Dp^N^* = 292; and for poly(l-Leu), *Dp^N^* = 1550. The average *M_n_^N^* were estimated as 184,000 for poly(l-Ala), 28,900 for poly(l-Val), and 175,000 for poly(l-Leu). On the other hand, the poly(Gly) (**8**) is highly cohesive and lesser soluble in common organic solvents, which hindered the molecular weight determination by *N*-terminus labeling. Therefore, the average molecular weight of the poly(Gly) (**8**) was determined on comparison with the limiting viscosity of poly(l-Val) as described in the Appendix A. The limiting viscosity of poly(Gly) (**8**) was 0.53 fold compared with the poly(l-Val). The value of limiting viscosity reflects the molecular weight of poly(Gly). The estimated *M_n_^V^* and *Dp^V^* of poly(Gly) were 15,100 and 502, respectively (Appendix A). 

### 3.2. Electrospinning

At first, the concentration of poly(amino acids) was optimized to produce continuous, homogeneous fibers. Most stable streams of the spinning jets were obtained for the pre-spun solutions of poly(amino acid)s/TFA at the following concentrations; poly(l-Ala) (**2**) = 12 wt.%, poly(l-Val) (**4**) = 12 wt.%, poly(l-Leu) (**6**) = 4 wt.%, and poly(Gly) (**8**) = 23–25 wt.%. The electrospinning conditions were further examined using pre-spun solutions of poly(amino acids) prepared in TFA:DCM = 10:0, 8:2, 5:5, and 2:8. The poly(amino acids) were insoluble in TFA:DCM = 2:8. 

The pre-spun solution of poly(l-Ala) in pure TFA was fabricated into poly(L-Ala)-ESNW, which was composed of fine fibers having average fiber diameter *φ* = 338 ± 50 nm (Figure 1a). Thinner fibers were obtained from the pre-spun solutions prepared in TFA:DCM = 8:2 and 5:5 which had average fiber diameters *φ* = 314 ± 77 nm (Figure 1b) and 278 ± 118 nm (Figure 1c), respectively. Similarly, poly(l-Val)- and poly(l-Leu)-ESNWs were prepared by the same spinning procedure for the poly(l-Ala)-ESNW. The average fiber diameters for poly(l-Val)-ESNWs are *φ* = 356 ± 175 nm (pure TFA, Figure 2a), 360 ± 249 nm (TFA:DCM = 8:2, Figure 2b), 369 ± 207 nm (TFA:DCM = 5:5, Figure 2c); for poly(l-Leu)-ESNWs *φ* = 338 ± 50 nm (pure TFA, Figure 3a), 316 ± 87 nm (TFA:DCM = 8:2, Figure 3b), 362 ± 119 nm (TFA:DCM = 5:5, Figure 3c). 

In the case of poly(Gly)-ESNW, the thinnest fibers, having average fiber diameters *φ* = 208 ± 108 nm (pure TFA, Figure 4a), 183 ± 115 nm (TFA:DCM = 8:2, Figure 4b), were obtained, but a small fraction of beads were also observed along the fiber axis. Despite the fact that the poly(Gly) have low molecular weight compared to other poly(amino acid)s, it gave fibers that have smaller diameters ranging from nano- to sub-micron scale. This phenomenon is attributed to the existence of strong cohesive forces that prevent solubility in the spinning solvent during electrospinning, which is the driving force for solidification of the thin fibers.

### 3.3. Analysis of Internal Structures in Poly(amino acid)-ESNWs

The wide-angle X-ray diffractogram of poly(l-Ala)-ESNWs are shown in Figure 5a. The poly(l-Ala)-ESNW prepared from pre-spun solution in TFA:DCM = 10:0 exhibited diffraction peaks at 2*θ* = 11.7°, 20.8°, and 23.8°. The poly(l-Ala)-ESNWs prepared from pre-spun solution in TFA:DCM = 8:2, 5:5 showed diffraction peaks at 2*θ* = 11.9°, 20.8°, and 23.8°. The intense diffraction peak observed at 2*θ* = 11.7^o^ (sum of Miller indices, S*_hkl_* = 1; d-spacing, d = 7.55 Å) for poly(l-Ala) (TFA:DCM = 10:0), indicated high degree of crystallinity and orientation in the fibers. The FT-IR spectroscopy for poly(l-Ala)-ESNWs exhibited amide I and amide II bands at 1654 cm^−1^ and 1550 cm^−1^ (Figure 5a, inset, curve 1), respectively, corresponding to an *α*-helical conformation [24]. Therefore, the most intense diffraction peaks at 2*θ* = 11.7° and 11.9° could be assigned to the *α*-helical chain conformation in the fibers. The low intense diffraction peaks at 2*θ* = 20.8° (S*_hkl_* = 3, d = 4.35 Å) and 23.8° (S*_hkl_* = 4, d = 3.77 Å) suggested for the presence of smaller fractions of chain conformations other than those ordered structure observed for the predominant peaks at 2*θ* = 11.7° and 11.9°. In FT-IR spectra, the overlapped adsorption bands at ca. 1638 cm^−1^ and 1539 cm^−1^, respectively, may be due to *β*-sheet conformation. Upon increasing the DCM content to 20% and 50%, the intensities of all the diffraction peaks increases, where no band shifts for amide I and amide II were observed in FT-IR spectra (Figure 5a, inset, curves 2 and 3), indicating that the *α*-helix content in the post-spun poly(l-Ala) fibers induce the higher crystallinity. 

The poly(l-Leu)-ESNW prepared from TFA:DCM = 10:0 solution exhibited two diffraction peaks at 2*θ* = 7.9° (S*_hkl_* = 1, d = 11.08 Å) and 18.3° (S*_hkl_* = 6, d = 4.58 Å) (Figure 5c). Similar to poly(l-Ala) fibers, for poly(l-Leu) fibers also predominant *α*-helical conformation was suggested by FT-IR spectroscopy, in which amide I and amide II bands were observed at 1650 cm^−1^ and 1543 cm^−1^ (Figure 5c, inset). The diffraction peaks for ordered structure in poly(l-Val)-ESNWs were observed at 2*θ* = 8.5° (S*_hkl_* = 1, d = 10.4 Å) and 19.0° (S*_hkl_* = 5, d = 4.67 Å) (Figure 5b). The poly(l-Val) molecules adopted mainly *β*-strand structure [18], which was insisted by the amide I and amide II bands at 1637 cm^−1^ and 1541 cm^−1^ (Figure 5b, inset). 

The WAXD pattern for poly(Gly)-ESNWs prepared in TFA:DCM = 10:0 and 8:2 showed diffraction peaks at 2*θ* = 11.9° (S*_hkl_* = 1, d = 7.33 Å) and 21.0° (S*_hkl_* = 3, d = 4.23 Å), respectively (Figure 5d). The ATR-IR spectra of poly(Gly)-ESNW (TFA:DCM = 10:0) exhibited amide I and amide II bands at 1650 cm^−1^ and 1544 cm^−1^ (Figure 5d, inset, curve 1) suggesting the random coil nature of the polymer along with a smaller fraction of poly(Gly)-II (PGII) structure [24]. The amide II band for poly(Gly)-ESNW prepared in TFA/DCM appeared at higher frequencies (1653 cm^−1^) than those prepared in pure TFA (1650 cm^−1^), which may contain a mixture of PGI, PGII, and amorphous structures.

Figure 5e illustrates the change in crystallinity of the uniaxially drawn poly(l-Ala)-film. The X-ray diffractogram of poly(l-Ala)-film before and after stretching showed peaks corresponding to an ordered structure at 2*θ* = 11.7° and 22.8° (Figure 5e). The amide I and amide II bands for the drawn poly(l-Ala)-film appeared at 1651 cm^−1^ and 1537 cm^−1^ (Figure 5e, inset, curve 1) indicated the existence of molecular chains in *α*-helical structure [24]. The pre-drawn film exhibited amide I and amide II bands at 1643 cm^−1^ and 1535 cm^−1^, respectively, for which the conformation was unidentifiable. The drawn film showed circular diffraction pattern (Appendix A) along with the drawing axis. This indicates that the molecular chains are oriented along with the drawing axis by the applied mechanical energy. The WAXD and FT-IR results for poly(l-Ala)-ESNWs and films insisted that the polymer molecules were mainly *α*-helices, though a smaller fraction of other conformations may exist.

### 3.4. Ordered Structure Content

The content of ordered structure (*C,* cps/mg) was determined based on the following equation (Equation (1)),
*C* = *I*_max_/*W*(1)
where *I*_max_ is the maximum diffraction intensity (cps), and *W* is the sample weight (mg) of the ESNW.

Table 1 summarizes the content of ordered structure transition due to the changes in the composition of the electrospinning solvent. Regardless of the poly(amino acids), for the ESNWs spun from pre-spun solutions having DCM composition more than 20%, the ordered structure content increased to 4–6-fold compared with the system using pure TFA.

In electrospinning process, the rapid solidification of the polymers due to the evaporation of the spinning solvent prevents the formation of highly ordered structures in the ESNWs. From the above discussion, it becomes clear that at any electrospinning conditions studied for poly(l-Ala), poly(l-Val), poly(l-Leu), and poly(Gly), the molecular chains form ordered secondary structures, which could be controlled by selection of different solvents. Tanioka et al. [25] has recently reported the highly oriented structures of electrospun PBLG fibers prepared from biphasic solutions. The results showed that the PBLG fibers had a uniaxially oriented structure, *α*-helices in their internal structures. Similarly, by employing appropriate spinning solvent and technique poly(amino acid)-ESNWs having highly oriented structures could be produced by conventional electrospinning technique.

## 4. Conclusions

A series of results presented in this study showed that the hydrophobic poly(*α*-amino acid)s, poly(l-Ala), poly(l-Leu), poly(l-Val), and poly(Gly) could be conveniently electrospun in TFA and TFA/DCM solvent mixtures. Poly(Gly) gave most thinnest fibers, which had average diameters in the range between 68–306 nm. The post-spun chain conformational analyses revealed that poly(l-Ala)- and poly(l-Leu)-fibers exhibit predominant *α*-helical structure, and poly(Gly) possessed random coil structures. Even as the preceding researches suggested that the electrospun poly(amino acid)-, polypeptide-fibers primarily adopted *α*-helical conformation, for the first time in this study, whereas poly(l-Val)-fibers possessed mainly the *β*-strand structure. The estimation of ordered structure content implied that upon careful selection of appropriate electrospinning solvent, the conformation of the molecular chains in the fibers could be controlled. These electrospun poly(amino acids)-ESNWs are expected to inspire novel applications of great potential from the view point of materials chemistry.

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
