# Peer review of "Electrospinning and Post-Spun Chain Conformations of Synthetic, Hydrophobic Poly(α-amino acid)s"

_polymers, 2020, doi:10.3390/polym12020327_

Round 1

Reviewer 1 Report

The research activity is interesting and fits well with the scopes of the journal.

By focusing on the electrospinning of Poly(α - amino acid)s, the paper analyses fibers conformations realized by using different solvent mixtures. An adequate number of data are reported and supported by references; moreover, paragraphs structure are well defined so the paper is clear and easy to read. Conclusions are consistent with both premises and discussion.

Author Response

Author's Notes to Reviewer_1

Q. The research activity is interesting and fits well with the scopes of the journal. By focusing on the electrospinning of Poly(α-amino acid)s, the paper analyses fibers conformations realized by using different solvent mixtures. An adequate number of data are reported and supported by references; moreover, paragraphs structure are well defined so the paper is clear and easy to read. Conclusions are consistent with both premises and discussion.

Ans. The authors are thankful for the valuable evaluation of the Reviewer #1 and acceptance for publication.

Reviewer 2 Report

Very good work and highly recommended for publication.

Author Response

Q. Very good work and highly recommended for publication.

Ans. The authors are thankful for the kind recommendation for publication by the Reviewer#2, who has understood the potential interests described in the original manuscript.

Reviewer 3 Report

A series of results presented in this study showed that the hydrophobic poly(-amino acid)s,

290 poly(L-Ala), poly(L-Leu), poly(L-Val), and poly(Gly) could be conveniently electrospun in TFA and291 TFA/DCM solvent mixtures. Poly(Gly) gave most thinnest fibers, which had average diameters in292 the range between 68–306 nm. The post-spun chain conformational analyses revealed that poly(L293Ala)- and poly(L-Leu)-fibers exhibit predominant -helical structure, and poly(Gly) possessed294 random coil structures. Even as the preceding researches suggested that the electrospun poly(amino295 acid)-, polypeptide-fibers primarily adopted -helical conformation [10,13,19], for the first time in296 this study, poly(L-Val)-fibers were found to posses mainly -strand structure. The estimation of297 ordered structure content implied that upon careful selection of appropriate electrospinning solvent,298 the conformation of the molecular chains in the fibers could be controlled. These electrospun299 poly(amino acids)-ESNWs are expected to inspire novel applications of great potential from the view300 point of materials chemistry.

In this manuscript (polymers-696323), authors prepared a series polyamino acids for electrospinning. Comprehensive investigations on the electrospinning conditions and the structure of polymers have been performed. In general, the work is well-designed, and the conclusion is supported by the experiment and results. However, there are still some minor issues to be addressed.

More details on the raw materials should be provided. More introduction on the purpose of this work should be clarified. Mechanical properties are important for the practical application of electrospun nanofibers. How about the mechanical performance of electrospun samples in this work? References are cited improper. Please do not cite references in conclusion part. In introduction part, more important, very recent and important references should be cited. For examples: Electrospinning of polyamino acid: Rapid Commun. 38(14) (2017) 1700147; J. Mater. Sci. 54(8) (2019) 6719–6727. Introduction of application of electrospun nonwovens: Filtration (Macromolecular Materials and Engineering, 2017, 302(1), 1600353-1600380; Macromolecular Materials and Engineering, 2018, 1800336-1800354); nanofibers reinforced composites (Compos. Sci. Technol. 174 (2019) 20-26; Polym. Chem. 9(20) (2018) 2685-2720.); bioengineering (https://doi.org/10.1016/j.cclet.2019.07.033); actuators (Macromol. Rapid Commun. 39(10) (2018) 1800082; Mater. Eng. 304(2) (2019) 1800548.); Chemosensors for Heavy Metal Detection (https://doi.org/10.1002/smll.201604293); etc. Etc. Why authors chose the solvent systems for electrospinning should be stated.

Author Response

Author's Notes to Reviewer_3

Responses from authors to Reviewer#3

The authors are thankful for the variable suggestions and comments from Reviewer #3 about revision of the original manuscript before acceptance for publication. All of the comments and suggestions have been incorporated in the revised manuscript, and the authors are hoping the manuscript has been satisfactory revised. The answers from the authors are listed below, which are corresponding to each of comments and questions from Reviewer#3.

Q1. More details on the raw materials should be provided.

Ans. The details on the raw materials including the synthetic procedures for precursors and their corresponding references are available in the Experimental Section in pages 4-8 of the revised manuscript. Also, Supplementary Information describes the characterization of the poly(amino acids) by means of chemical method to estimate the degree of polymerization.

Q2. More introduction on the purpose of this work should be clarified.

Ans. The purpose and objectives of the present study are available in page 4 of the revised manuscript, as well as a reference [18] was newly added to introduce a preceding work on the electrospinning of a low molecular weight poly(L-Phe) preparation.

Q3. Mechanical properties are important for the practical application of electrospun nanofibers. How about the mechanical performance of electrospun samples in this work?

Ans. The authors are thankful for the prospective comment by the Reviewer#3, because the Q3 is highly suggestive for our works ongoing towards the application of the poly(amino acids) ESNWs. For an instance as typically known for silk and wool, the chain conformation of the polypeptide (fibrous proteins; fibroin for silk and keratin for wools) is a major factor affecting the mechanical properties of the natural proteinaceous fibers.
So that the relation between the adaptable chain conformation of poly(amino acids) and the mechanical property of the electrospun fibers are of our interests, as the Reviewer#3 has pointed out. On the other hands, the actual methodology of the mechanical evaluation in the case of ESNWs, however, is not a straightforward one using a known, routine "uniaxially drawing test (or tensile test)" on a single filament or a thread of staple fibers (yarns), because the electrospun poly(amino acids) fibers comprise of the non-woven matrices, which means that, for uniaxially drawing tests, a single filament cannot be physically separated from the non-woven fabrics, as well as, if a tape (or a narrow ribbon)-like shape of the ESNW samples is tested, the stress-strain (or load-displacement) curve will not reflect the mechanical property of the single poly(amino acids) filament but of the non-woven.
According to the reason described above, the methodology of the mechanical evaluation of poly(amino acids)-ESNWs should be carefully established, in order to discuss the effect of adaptable chain conformation on the required physical properties for application. And this is also the reason why the authors would like to concentrate just for giving the readers a successful preparation method of ESNW, combined with their adaptable chain conformation.
The author hopes that Revewer#3 kindly understands that the main objective of this work is to study the post-spun chain conformation of electrospun poly(alpha-amino acids), as entitled. The works on the mechanical evaluation will be reported as a separated research article in future.

Q4. References are cited improper. Please do not cite references in conclusion part.

Ans. The citations in the conclusion section have been removed as suggested.

Q5. In introduction part, more important, very recent and important references should be cited. For examples: Electrospinning of polyamino acid: Rapid Commun. 38(14) (2017) 1700147; J. Mater. Sci. 54(8) (2019) 6719–6727. Introduction of application of electrospun nonwovens: Filtration (Macromolecular Materials and Engineering, 2017, 302(1), 1600353-1600380; Macromolecular Materials and Engineering, 2018, 1800336-1800354); nanofibers reinforced composites (Compos. Sci. Technol. 174 (2019) 20-26; Polym. Chem. 9(20) (2018) 2685-2720.); bioengineering (https://doi.org/10.1016/j.cclet.2019.07.033); actuators (Macromol. Rapid Commun. 39(10) (2018) 1800082; Mater. Eng. 304(2) (2019) 1800548.); Chemosensors for Heavy Metal Detection (https://doi.org/10.1002/smll.201604293); etc. Etc.

Ans. As suggested by the Reviewer#3, suitable references are included in the revised manuscript. The authors have read through and carefully reviewed the contents and descriptions on totally ten references suggested by Reviewer#3, and among them, the authors have find that five references potentially involve the relevance to our original manuscript.

The five references listed below, hence, were newly cited in the revised manuscript:

10. Tingting, W.; Mengzhen, D.; Cuiping, S.; Yiqun, Q.; Panpan, W.; Ruirui, Q.; Xichang, W.; Jian, Z. Resorbable polymer electrospun nanofibers: History, shapes and application for tissue engineering. Chin. Chem. Lett. 2019, accepted, in press.
11. Miaomiao, Z.; Jingquan, H.; Fang, W.; Wei, S.; Ranhua, X.; Qilu, Z.; Hui, P.; Yong, Y.; Sangram, K.S.; Feng, Z.; Chaobo, H. Electrospun nanofibers membranes for effective air filtration. Macromol. Mater. Eng. 2017, 302, 1600353.
12. Dan, L.; Miaomiao, Z.; Zhicheng, J.; Shaohua, J.; Qilu, Z.; Ranhua, X.; Chaobo, H. Green electrospun nanofibers and their application in air filtration. Macromol. Mater. Eng. 2018, 303, 1800336.
13. Li, L.; Hadi, B.; Shaohua, J.; Holger, S.; Seema, A. Composite polymeric membranes with directionally embedded fibers for controlled dual actuation. Macromol. Rapid Commun. 2018, 39, 1800082.
14. Nan, Z.; Ruirui, Q.; Jing, S.; Juan, Y.; Zhiqiang, X.; Yiqun, Q.; Xichang, W.; Jian, Z. Recent advances of electrospun nanofibrous membranes in the development of chemosensors for heavy metal detection. Small 2017, 13, 1604293.

In addition to above references, we have found one to be added in the revision:

19. Hiroaki, Y., Kazuhiro, Y. Creation of superhydrophobic poly(L-phenylalanine) nonwovens by electrospinning. Polym. 2018, 10, 1212.

Q5. Why authors chose the solvent systems for electrospinning should be stated.

Ans. Our research group has examined the chain conformations of the poly(gamma-benzyl-L-glutamate) (PBLG) in pre- and post-electrospinning process. The PBLG was electrospun in dichloromethane (DCM) and DCM/trifluoroacetic acid (TFA). Even as the pre-spun solution in TFA or DCM/TFA exhibited random chain conformation, the molecular chains in post-spun PBLG fibers showed alpha-helical structure. Hence, the authors selected the same solvent system for electrospinning of hydrophobic poly(alpha-amino acid)s, which allows the authors to compare the post-spun chain conformations of previous PBLG ESNW and the present series of hydrophobic poly(amino acids). The corresponding section is available in page 3 of the revised manuscript.